# Polycaprolactone-Based Scaffolds Facilitates Osteogenic Differentiation of Human Adipose-Derived Stem Cells in a Co-Culture System

**DOI:** 10.3390/polym13040597

**Published:** 2021-02-17

**Authors:** Ismail Rozila, Pedram Azari, Sha’ban Munirah, Wan Kamarul Zaman Wan Safwani, Belinda Pingguan-Murphy, Kien Hui Chua

**Affiliations:** 1Department of Physiology, Faculty of Medicine, University Kebangsaan Malaysia, Kuala Lumpur 56000, Malaysia; nurhanisraihanah@gmail.com; 2Department of Pharmaceutical Sciences, Faculty of Pharmacy, University of Cyberjaya, Selangor 63000, Malaysia; 3Department of Chemistry, Faculty of Science, University of Malaya, Kuala Lumpur 50603, Malaysia; Pedram.azari@gmail.com; 4Department of Rehabilitation and Physiotherapy, Kulliyyah of Allied Health Sciences, International Islamic University Malaysia, Kuantan 25200, Malaysia; munirahshaban@iium.edu.my; 5Department of Biomedical Engineering, Faculty of Engineering, University of Malaya, Kuala Lumpur 50603, Malaysia; wansafwani@um.edu.my (W.K.Z.W.S.); bpingguan@um.edu.my (B.P.-M.)

**Keywords:** human adipose-derived stem cells, osteogenic, polycaprolactone

## Abstract

(1) Background: Stem cells in combination with scaffolds and bioactive molecules have made significant contributions to the regeneration of damaged bone tissues. A co-culture system can be effective in enhancing the proliferation rate and osteogenic differentiation of the stem cells. Hence, the aim of this study was to investigate the osteogenic differentiation of human adipose derived stem cells when co-cultured with human osteoblasts and seeded on polycaprolactone (PCL):hydroxyapatite (HA) scaffold; (2) Methods: Human adipose-derived stem cells (ASC) and human osteoblasts (HOB) were seeded in three different ratios of 1:2, 1:2 and 2:1 in the PCL-HA scaffolds. The osteogenic differentiation ability was evaluated based on cell morphology, proliferation rate, alkaline phosphatase (ALP) activity, calcium deposition and osteogenic genes expression levels using quantitative RT-PCR; (3) Results: The co-cultured of ASC/HOB in ratio 2:1 seeded on the PCL-HA scaffolds showed the most positive osteogenic differentiation as compared to other groups, which resulted in higher ALP activity, calcium deposition and osteogenic genes expression, particularly Runx, ALP and BSP. These genes indicate that the co-cultured ASC/HOB seeded on PCL-HA was at the early stage of osteogenic development; (4) Conclusions: The combination of co-culture system (ASC/HOB) and PCL-HA scaffolds promote osteogenic differentiation and early bone formation.

## 1. Introduction

Stem cells have been found to have significant impacts on regenerative medicine and tissue engineering. Due to their self-renewal and differentiation ability, stem cells have been seen as an alternative or supplementary to the current existing medical treatment of many diseases, including bone disease. Bone-related disease treatment choice is limited due to its low regenerative capacity which leads to patients going for either surgery or seek symptomatic treatment and preventive measures are recommended for those who are susceptible, particularly for osteoarthritis cases [1]. However, reliance on these options may fail to address the underlying problems regarding bone disease. An ideal treatment for bone disease—or any disease for that matter—should have optimum effects, be less invasive and have less side-effects to the patients. Stem cells may be the answer to this, and they have been shown to be successful in the treatment of bone-related diseases [2].

Although stem cells have been shown to have self-renewal and differentiation capacity, they need to be in an appropriate environment with the use of selected biomaterials to maneuver them into the correct differentiation pathway. In the case of osteogenic differentiation, the combination of stem cells and 3D scaffolds has been the hallmark of fabricating functional bone constructs [3,4,5]. The synthetic polymers such as polycaprolactone (PCL) are the most cost-effective materials, but exhibit the disadvantage that cell adhesion on the surface is reduced due to the strong hydrophobicity of the surface [6]. Therefore, PCL can be blended with a calcium apatite such as hydroxyapatite (HA), which increases the surface hydrophilicity and allow cell attachment, proliferation and differentiation of bone cells [7]. Scaffold materials such as hydroxyapatite (HA) in various formats have been shown to aid stem cells in osteogenic differentiation and bone formation [8,9]. Similarly, the composition of biomaterial consists of HA and PCL has also been shown to support bone tissue regeneration, being porous and biodegradable. These characteristics demonstrated that PCL/HA scaffolds may support attachment, proliferation and, hence, bone formation [10,11,12]. Apart from that, the hybridization of PCL and HA in bone tissue construct has been shown to overcome the inflexibility and brittleness that is normally associated with hard ceramic materials [13].

In addition to the influence of biomaterials composition, the paracrine activity of stem cell and osteoblasts in co-culture plays an important role as it can alter the bioactive components that are being activated in the event of bone tissue regeneration. This can be achieved through the co-culture system, which applies the fundamentals of cell to cell interaction and requires the cells to be in specific proximity in order to have an impact on cells response [14]. Therefore, this study aims to investigate the osteogenic differentiation potential using a co-culture system, in which we co-cultured human adipose-derived stem cells (ASC) and human osteoblast cells (HOB) seeded in hybrid composite of electrospun fibrous scaffolds material consisting of PCL and HA.

## 2. Materials and Methods

### 2.1. Human Adipose-Derived Stem Cells (ASCs) Isolation and Culture

ASCs were harvested from subcutaneous adipose tissue of female donors who underwent elective segmental abdominal Caesarean procedure at Universiti Kebangsaan Malaysia Medical Centre after obtaining informed consent from the patients. The study protocol was approved by Universiti Kebangsaan Malaysia Research and Ethical Committee (Approval code of FF-2015-247). The adipose tissue was digested using 0.3% Collagenase Type I (Sigma-Aldrich, St. Louis, MO, USA) for 2 h at 37 °C and which was then centrifuged at 200× *g* for 10 min. The supernatant was then discarded and the cell pellet was washed with phosphate buffered saline (PBS) twice. The cells were cultured in Dulbecco’s Modified Eagle Medium (DMEM)/HamF12 medium supplemented with 10% fetal bovine serum (FBS; Invitrogen, Carlsbad, CA, USA), 1% antibiotic–antimycotic (Invitrogen, Carlsbad, CA, USA), 1% glutamax (Invitrogen, Carlsbad, CA, USA) and 1% vitamin C (Sigma-Aldrich, St. Louis, MO, USA). Cell cultures were incubated and maintained at 37 °C with 5% carbon dioxide and were fed every 3 days. The cells were grown to confluence at 80–90% after the initial plating (*p* = 0) and they were sub-cultured at 1:4 expansion under the same condition until passage 4 (P4). Once confluent, the adherent cells were detached using 0.125% trypsin–EDTA. ASC passage 4 (P4) was used for testing and analysis. Microscopic examination was done to confirm stem cell characteristics by observation of fibroblast-like morphology and cell plastic adherence properties from passage 0 until passage 4 prior cell seeding.

### 2.2. Human Osteoblasts (HOB) Isolation and Culture

Human osteoblasts were isolated from the nasal bone harvested from patients who underwent elective septoplasty surgery at the Universiti Kebangsaan Malaysia Medical Centre after obtaining informed consent. This study protocol was approved by the Universiti Kebangsaan Malaysia Ethics and Research Committee (Approval code of FF-2015-247). The nasal bone was cut into small pieces of 1 mm^3^ and washed using phosphate buffered saline (PBS; Gibco, NY, USA), after which they were digested using 0.6% Collagenase Type I (Sigma-Aldrich, St. Louis, MO, USA) for 2 h at 37 °C. After digestion, the isolated osteoblast was pelleted down by centrifugation at 600× *g* for 10 min. The osteoblasts were then cultured in Dulbecco’s Modified Eagle Medium (DMEM):Ham’s F12 medium (Gibco; 1:1) supplemented with 10% fetal bovine serum (FBS; Gibco), 1% antibiotic–antimycotic (Invitrogen, Carlsbad, CA, USA), 1% glutamax (Invitrogen) and 1% vitamin C (Sigma-Aldrich). They were incubated and maintained at 37 °C with 5% carbon dioxide (CO_2_). Culture medium was changed every 3 days. When the cells reached confluence at 80–90% after initial plating (P0), the cells were then sub-cultured at 1:4 expansion ratios under the same condition until passage 2 (P2). Detachment of cells for sub-culturing purpose was carried out using 0.125% trypsin–EDTA (Gibco). HOBs at P2 were used for testing and analysis. Microscopic examination was done to confirm cell morphology of osteoblast remained a cuboidal-like shape and polygonal –like from passage 0 until passage 2 before cell seeding into PCL-HA scaffold.

### 2.3. PCL-HA Scaffolds Preparation and Fabrication

Bovine HA (BHA) powder was prepared from bovine bone which was obtained from a local slaughterhouse. Bovine bone and PCL (Mn = 80,000 gmol-1; Aldrich) were first processed individually according to the methods described elsewhere [9]. Solution blending of PCL and BHA was carried out in weight compositional ratios of 50:50. PCL was dissolved in a solvent containing nine parts of chloroform (Merck, Kenilworth, NJ, USA) and one part of dimethylformamide (Friendemann Schmidt, Diviney Court Parkwood, WA, USA) at 50 °C for approximately 3 h until the mixture became homogenous before BHA was added after which the stirring continued for another 20 h. The mixture was then sonicated for 1 h to prevent the agglomeration of BHA. The electrospun scaffold consists of PCL and BHA was fabricated and characterized according to the previous experimental set-up [10]. The electrospun of PCL-HA were then cut into disc shapes (diameter = 5 mm and thickness = 698 µm) before use.

### 2.4. Co-Culture Set-Up

Three co-cultured groups consisting of ASC and HOB were prepared in different ratios, ASC/HOB: (1) 1:1; (2) 2:1; and (3) 1:2. All groups were seeded onto the electrospun PCL-HA scaffolds with approximately 50,000 cells of each scaffold. The co-cultured cells were cultured and maintain in a culture condition as describe in Section 2.1 and Section 2.2. Meanwhile, monocultures of ASC and HOB seeded separately onto electrospun PCL-HA were used as controls. All co-cultured groups were evaluated for morphology, growth and osteogenic differentiation ability and compared with the controls.

### 2.5. Morphology and Cell Proliferation Analysis

The morphological analysis of the controls and the co-cultured groups within the PCL-HA scaffold were analyzed microscopically using Field Emission Scanning Electron Microscopy (FESEM) (JEOL, JSM 6700F model) at Day 1, 7 and 14. Before microscopic analysis, all cell samples were washed with PBS and fixed with 4% paraformaldehyde (Sigma Aldrich) for 1 h and subsequently processed in graded series of alcohol before being dried overnight at room temperature. All samples were sputter coated with gold prior to observation using FESEM.

Cell proliferation ability of the cell samples were also analyzed using AlamarBlue assay (Invitrogen, Carlsbad, CA, USA) according to manufacturer’s protocol at days 7 and 14. Cells were seeded onto a 24 well-plate at approximately 5 × 10^4^ cells per well. Absorbance signals were quantified at excitation wavelength of 570nm and emission wavelength of 595 nm using a microplate reader (FLUOstar Optima, BMG Labtech, Offenburg, Germany). Viable cells were quantified as percentage of resazurin reduction according to the pre-equilibration standard curve. A bar graph was then plotted by potting the cell numbers against time (day) to compare the growth rate of cells.

### 2.6. Alkaline Phosphatase (ALP) Analysis

QuantiChrom ALP kit (BioAssay Systems, Hayward, CA, USA) was used to evaluate the ALP activity of controls and co-cultured groups seeded in the PCL-HA scaffolds at days 7 and 14, in accordance with the manufacturer’s recommendation. The absorbance of all cell samples was analyzed at the wavelength of 405 nm using a microplate reader (ThermoFisher Scientific, Multiskan^TM^ GO model, Vantaa, Finland).

### 2.7. Calcium Deposition Analysis

Calcium deposition of the controls and co-cultured groups seeded in the PCL-HA scaffolds was evaluated using Alizarin Red S (ARS) staining kit (Merck) at days 7 and 14 according to manufacturer’s protocol. The absorbance of the cell samples was analyzed ay 405 nm wavelength using a microplate reader (Thermo Scientific, Multiskan^TM^ GO model) after which the amount of calcium deposition was analyzed according to the standard curve obtained from pre-equilibration process. The stained cell samples were also microscopically observed using a microscope (Nikon SMZ645).

### 2.8. Quantitative Real-Time Polymerase Chain Reaction (RT-PCR)

Total RNA was extracted using TRI reagent (Molecular Research Center, Cincinati, OH, USA). The extraction process was carried in accordance with the manufacturer’s protocol. The yield and purity of the total RNA was determined using a spectrophotometer (Bio-Rad, Hercules, CA, USA). The total RNA was then used to synthesize complementary DNA (cDNA) using SuperScript III reverse transcriptase (Invitrogen) and procedure was carried out according to manufacturer’s instructions. The protocol conditions for cDNA synthesis were 10 min at 23 °C for primer annealing, 60 min at 42 °C for reverse transcription and 10 min at 94 °C for reaction termination.

The cDNA was then used as a template for RT-PCR to evaluate the gene expression level of osteogenic-associated genes: Runx2, osteopontin (OSP), osteocalcin (OCN) and bone sialoprotein (BSP) and *alkaline phosphatase* (ALP); in the controls and the co-cultured groups. All genes were normalized to *glyceradehyde-3-phosphate dehydrogenase* (GAPDH), which is a housekeeping gene. Primers for all genes were designed using Primer 3 software based on the published GeneBank database sequences (Table 1). SYBR Green was used as an indicator during PCR reaction in BioRad iCycler PCR machine. The reaction conditions for PCR were: cycle 1: 95 °C for 3 min (1×); cycle 2: Step 1: 95 °C for 10 s; and Step 2: 61 °C for 30 s (40×). Each primer used for PCR were then confirmed for specificity with melting curve analysis.

### 2.9. Statistical Analysis

Quantitative data from the cell growth, ALP, cell mineralization and gene expression levels are expressed as mean ± standard error of mean (SEM). Student T-test (IBM SPSS Statistics 20) was carried out to determine statistical significance of all cell samples tested. A *p* value of < 0.05 was statistically significant.

## 3. Results

### 3.1. Cell Morphology in PCL-HA

From the FESEM images, we observed HOB and ASC adhesion at the surfaces of the PCL-HA scaffold at days 7 and 14. The HOB showed a few colonies with oval morphology at day 7 (Figure 1A) as compared to 14 (Figure 1B), which they have extended to spread more homogenously with some appearance of extracellular matrix.

ASC seems to attach to the surfaces of the PCL-HA scaffold at day 7 (Figure 1C). The number of cells appeared to be increased at day 14 (Figure 1D) with some appearance of fibroblastic morphology. The co-cultured group consisting of ASC/HOB with a ratio of 1:1 exhibited good adhesion to the microfibers at day 7 (Figure 1E), although they did not appear to be homogenous as compared to day 14 (Figure 1F), at which the cells appeared to be more polygonal and homogenous. Similar features can be observed in co-cultured group with the ratio of 2:1 (Figure 1G) at day 7 and day 14 (Figure 1H), which the cells appeared to be more homogenous with the formation cell projection with better adhesion and growth. The co-cultured group consisting of ASC/HOB with the ratio of 1:2 seems to show better cell adhesion and penetration inside the scaffold at day 7 (Figure 1I). While at day 14, cells appeared to be more homogenous and more polygonal in shape, indicating osteoblastic characteristics (Figure 1J).

### 3.2. Cell Proliferation Ability in PCL-HA Increased with Co-Cultured ASC/HOB at 2:1 Ratio

To determine the cell proliferation ability of the cell samples, we measured the proliferation rate at days 1, 7 and 14 (Figure 2) using AlamarBlue assay. All groups showed significant increase in cell growth (*p* < 0.05) at day 14 compared to days 1 and 7. We found that co-cultured ASC/HOB at ratio of 2:1 showed a significant increase (*p* < 0.05) by 1-fold in cell growth at day 14 (4.2 × 10^5^ ± 8.45 × 10^3^) compared to the control groups, ASC (3.7 × 10^5^ ± 8.5 × 10^3^) and HOB (3.8 × 10^5^ ± 6.5 × 10^3^). Similarly, they were also found to be significantly increased (*p* < 0.05) by 1.2-fold when compared to other co-cultured groups ASC/HOB at ratio of 1:1 (3.4 × 10^5^ ± 4.2 × 10^3^) and 1:2 (3.5 × 10^5^ ± 1.6 × 10^3^).

### 3.3. ALP Activity Increased in Co-Cultured ASC/HOB Seeded in PCL-HA Scaffold

All groups were subjected to ALP activity evaluation at days 7 and 14 (Figure 3). ALP activity at day 14 was significantly higher (*p* < 0.05) as compared to day 7. Being a mature osteoblast, HOB (2.24 ± 0.08) has the highest ALP activity compared to other groups at day 14. Although there was an increase in ALP activity among the co-cultured groups at day 14, they were significantly lower (*p* < 0.05) as compared to HOB by approximately 1-fold. However, the co-cultured groups were significantly increased (*p* < 0.05) as compared to ASC by 3-fold in both ASC/HOB at 2:1 and 1:1 and by 2.5-fold in ASC/HOB at 1:2. On the other hand, there were no significant differences in the ALP activity among the co-cultured groups.

### 3.4. Calcium Deposition Was Enhanced in Co-Cultured ASC/HOB Seeded at 2:1 in PCL-HA Scaffold

All groups were subjected to calcium deposition evaluation (Figure 4) at days 7 and 14. Calcium deposition was found to be significantly increased (*p* < 0.05) at day 14 as compared to day 7 in HOB and in ASC/HOB co-cultured at 2:1, but it was significantly decreased (*p* < 0.05) in ASC. All groups except for ASC/HOB co-cultured at 1:1 was significantly increased (*p* < 0.05) when compared to ASC by approximately 3-fold in ASC/HOB co-cultured at 2:1 and 1:2 and by 2-fold in HOB. Among the co-cultured groups, we found that calcium deposition was significantly increased (*p* < 0.05) by approximately 3-fold in ASC/HOB co-cultured at 2:1 and 1:2 as compared those co-cultured at 1:1. Cell samples stained with Alizarin Red S was also observed microscopically (Figure 5) and results obtained reflects the data in Figure 4, where significant difference can be observed at day 14, particularly in ASC/HOB co-cultured at 2:1.

### 3.5. Alteration of Gene Expression in Cells Seeded in PCL-HA Scaffold

All groups were evaluated for osteogenic gene expression analysis when seeded in PCL-HA scaffold using quantitative RT-PCR (Figure 6). The analysis showed that co-cultured ASC/HOB at 2:1 showed the highest expression (*p* < 0.05) of Runx (6.45 × 10^−1^ ± 3.56 × 10^−2^) with more than 100 folds increase as compared to other groups at day 14 (Figure 6A). OSP showed a significant increase (*p* < 0.05) in expression in ASC/HOB co-cultured at 2:1 (5.35 × 10^−2^ ± 6.57 × 10^−4^) with more than 100 folds increase when compared to those co-cultured at 1:1 and 1:2 at day 14 (Figure 6B). However, OSP expression in ASC/HOB co-cultured at 2:1 was significantly lower (*p* < 0.05) when compared to HOB (3.39 × 10^−1^ ± 2.23 × 10^−2^). ALP expression (Figure 6C) was also significantly increased in ASC/HOB co-cultured at 2:1 (7.15 × 10^−2^ × 5.44 × 10^−3^) with more than 100 folds increase as compared to other groups but was significantly lower as compared to HOB (8.46 × 10^−2^ ± 1.48 × 10^−2^). Similarly, BSP expression was significantly (*p* < 0.05) increased in ASC/HOB co-cultured at 2:1 (1.29 × 10^−1^ ± 7.6 × 10^−3^) with more than 100 fold as compared to ASC, co-cultured groups at ratio of 1:1 and 1:2 (Figure 6D). OCN expression (Figure 6E) was significantly increased in ASC/HOB co-cultured at 2:1 (1.27 × 10^−1^ ± 2.52 × 10^−3^) when compared to those co-cultured at 1:1 and 1:2 with more than 50 folds and more than 100 folds increase, respectively but was significantly lower when compared to HOB (4.1 × 10^−1^ ± 2.78 × 10^−2^).

## 4. Discussion

In the present study, the effects of co-cultured ASC and HOB in the ratio of 2:1, 1:2 and 1:1 seeded in PCL- HA electrospun scaffold on cell growth, cell interactions and mineralization, ALP activity as well as osteogenic differentiation ability were evaluated. In this study, we employed the co-culture system to evaluate cell-cell interaction, specifically the interaction between ASC and HOB and its influence on osteogenic differentiation without the use of differentiation media. Based on the morphological and cell growth analysis, the cells response well to the 3D microenvironment fabricated using PCL-HA composites in which the cells were observed to adhere and attached to the scaffolds and the cell density increased from day 7 to day 14. For the co-cultured groups, ASC/HOB in the ratio of 2:1 showed the most positive enhancement in terms of cell growth and differentiation into osteogenic-like cells with its polygonal morphology as observed through microscopic visualization. This indicates that paracrine activity of co-cultured ASC/HOB in the correct ratio increase the ASC ability to differentiate into osteogenic cells and this may be enhanced by the use of PCL-HA composites, particularly HA [15] which has been shown to support bone formation as compared to the use of PCL alone [16] as HA has been reported to induce greater surface roughness, which increase surface area resulting in more cells attachment particularly suitable for differentiation along osteogenic lineage. In addition, surface roughness has been shown to influence the regulation of osteogenic differentiation [17].

The enhanced support towards osteogenesis using co-culture system and PCL-HA scaffold was reflected in the ALP activity which was observed to be increased in all of the co-cultured groups as compared to ASC. The activity of ALP was progressive over time as the culture period increased from day 7 to day 14, which was similarly reported in a variety of osteogenic mineralizing media conditions [18]. The enhancement of osteogenesis was also observed in cell mineralization analysis of co-cultured cells particularly in ASC/HOB co-cultured at 2:1, which showed an increase of calcium concentration from day 7 to day 14. Comparatively, the calcium concentration was even higher than the HOB and those co-cultured at 1:1 and 1:2 at day 14. This also indicates that ASC/HOB co-cultured at 2:1 has more positive effects in terms of osteogenic differentiation. This was further supported by the observation made from Alizarin Red S staining procedure, which showed that ASC/HOB co-cultured at 2:1 has a more concentrated calcium deposition as indicated by darker shade of red as compared to other groups. Calcium deposition or mineralization is one of the hallmarks of osteogenic differentiation [19] and the use of HA embedded scaffolds may produce more minerals deposits [20], which appeared to be more concentrated with the right combination of ASC/HOB (ratio 2:1) in the co-culture system. Moreover, the optimal co-culture system together with bioactive calcium from HA release into the microenvironment promote the calcium to be adsorbed and combined with the secreting proteins from the cells, thus induced ASC/HOB to attach, migrate and differentiate leading to osteogenic differentiation [21].

Analysis of osteogenic genes expression carried out using quantitative RT-PCR method showed that ASC/HOB co-cultured at ratio of 2:1 showed the highest osteogenic genes expression as compared to those co-cultured at ratio of 1:1 and 1:2. This indicates that 2:1 ratio seeded in the PCL- HA scaffold has the most positive effects on the osteogenic differentiation. ASC/HOB co-cultured at ratio of 2:1 appeared to have high expression of the osteogenic genes except for OSP and OCN, which are comparatively lower than Runx, ALP and BSP. OSP, a middle stage indicator for osteogenic differentiation is non-collagenase matrix protein that is important in stabilizing the main structural component of bone matrix [22], while OCN is a late stage indicator for osteoblast differentiation, which plays a role in the onset of matrix mineralization [23]. Runx, BSP and ALP are known to be early markers of the osteoblast differentiation [24]. These osteogenic markers have been found to have differential pattern of expression in long-term culture [25]. In the present study, the pattern of gene expression indicates that the co-cultured ASC/HOB, particularly for those seeded at 2:1 ratio maybe in the early stage of osteogenic development [26]. On the other hand, we can also assume that the ASC in the co-cultured condition may be experiencing a progressive transition in osteogenic development, whereby the cells may be moving from the early stage of development to the middle stage of development as analysis of cell mineralization indicates that some of the co-cultured cells population may already be in the middle stage of development or even in the midst of entering the late stage of development [27]. We believe that by increasing the culture period, more cells in the co-cultured groups particularly ASC/HOB at the ratio 2:1 may be able to enter the late stage of osteogenic development. This has been demonstrated using polygolic acid scaffolds, whereby increase of calcium concentration is contributed by the increase in time and cell density in the scaffold [28].

## 5. Conclusions

Upon osteogenic differentiation, ASC in the co-cultured groups assumed a more polygonal morphology of osteoblasts on the PCL- HA scaffolds, as confirmed by proliferation, ALP activity, cell mineralization and the osteogenic gene expression analysis. Under the current prescribed condition, the cells seeded in PCL- HA may be in the early stage of osteogenic differentiation. Coupled with the correct ratio of ASC and HOB in co-culture system, PCL-HA composite used in this study enhanced the osteogenic differentiation, which may be due to the increase in surface area and mechanical strength that has been known to support bone formation.

## Figures and Tables

**Figure 1 polymers-13-00597-f001:**
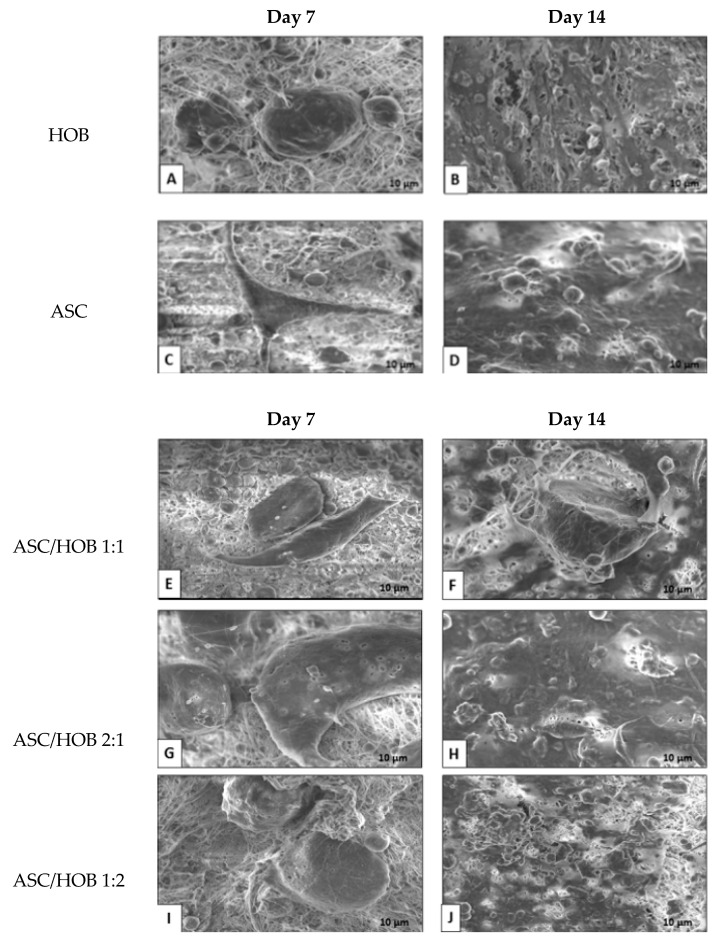
FESEM-images of cross sections of cell morphology in PCL-HA at days 7 and 14. Representative FESEM images of control groups (**A**,**B**) HOB and (**C**,**D**) ASC. panel; (**E**,**F**), ASC/HOB co-cultured at 1:1, (**G**,**H**) ASC/HOB co-cultured at 2:1 and (**I**,**J**) ASC/HOB co-cultured at 1:2.

**Figure 2 polymers-13-00597-f002:**
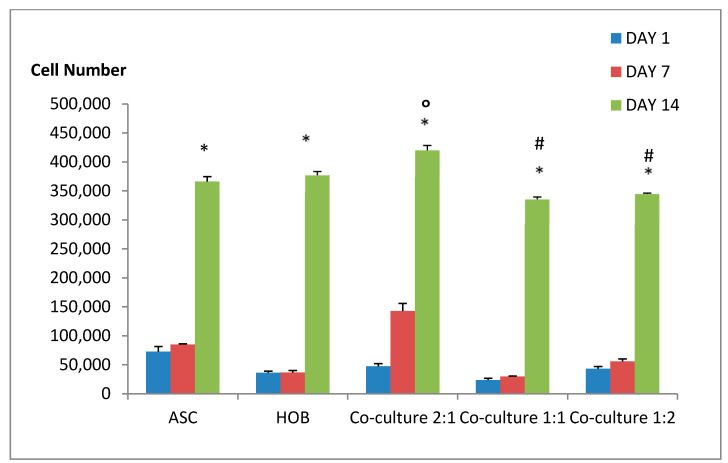
Cell growth in PCL-HA at days 1, 7 and 14. Most groups showed an increase in cell growth at day 14 at which ASC/HOB co-cultured at 2:1 showed the most significant increase as compared to other groups. * Indicates significant difference where *p* < 0.05, relative to days 1 and 7, # indicates significant difference where *p* < 0.05, relative to ASC/HOB co-cultured at ratio of 2:1 and O indicates significant difference where *p* < 0.05, relative to the control groups (ASC and HOB).

**Figure 3 polymers-13-00597-f003:**
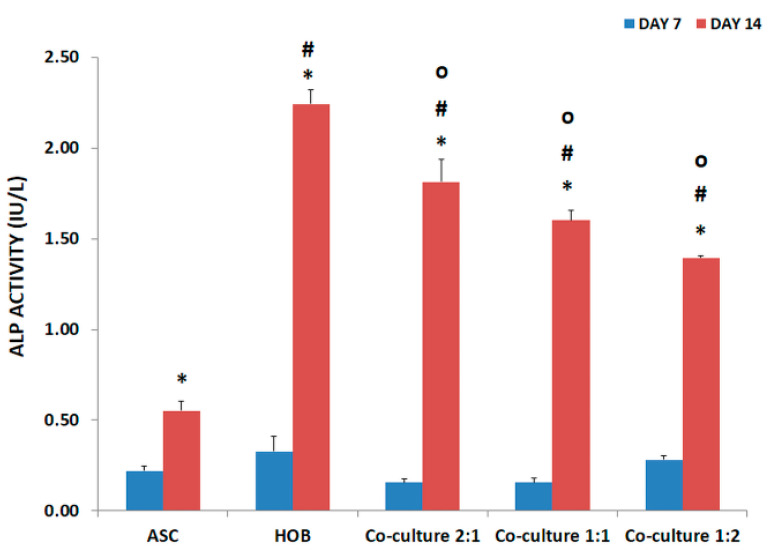
ALP activity in PCL-HA at days 7 and 14. ALP activity is the highest in HOB compared to other groups. ALP activity increased significantly in all co-cultured groups as compared to ASC at day 14. * Indicates significant difference where *p* < 0.05, relative to day 7, # indicates significant difference where *p* < 0.05, relative to ASC (control group), O indicates significant difference where *p* < 0.05, relative to HOB (control group).

**Figure 4 polymers-13-00597-f004:**
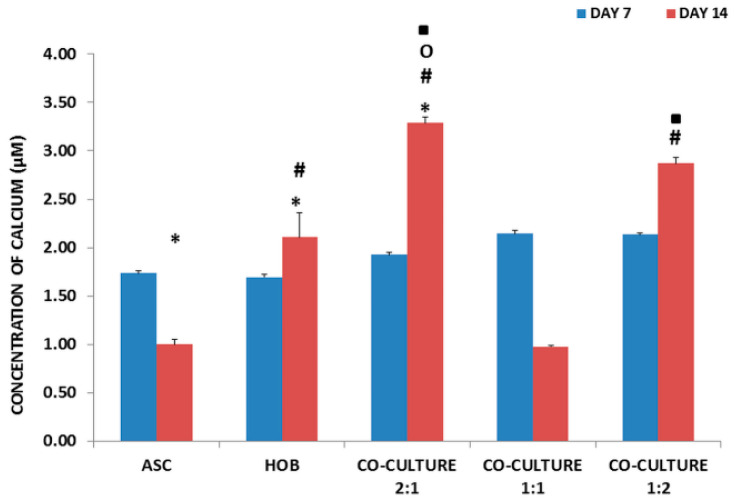
Calcium deposition in PCL-HA at days 7 and 14. Among the co-cultured groups, ASC/HOB at 2:1 has the highest calcium deposition (*p* < 0.05) and it is also significantly (*p* < 0.05) increased when compared to the control groups. * Indicates significant difference of *p* < 0.05, relative to day 7, # indicates significant difference of *p* < 0.05, relative to ASC, ^O^ indicates significant difference of *p* < 0.05, relative to HOB, ▪ indicates significant difference of *p* < 0.05, relative to ASC/HOB co-cultured at 1:1.

**Figure 5 polymers-13-00597-f005:**
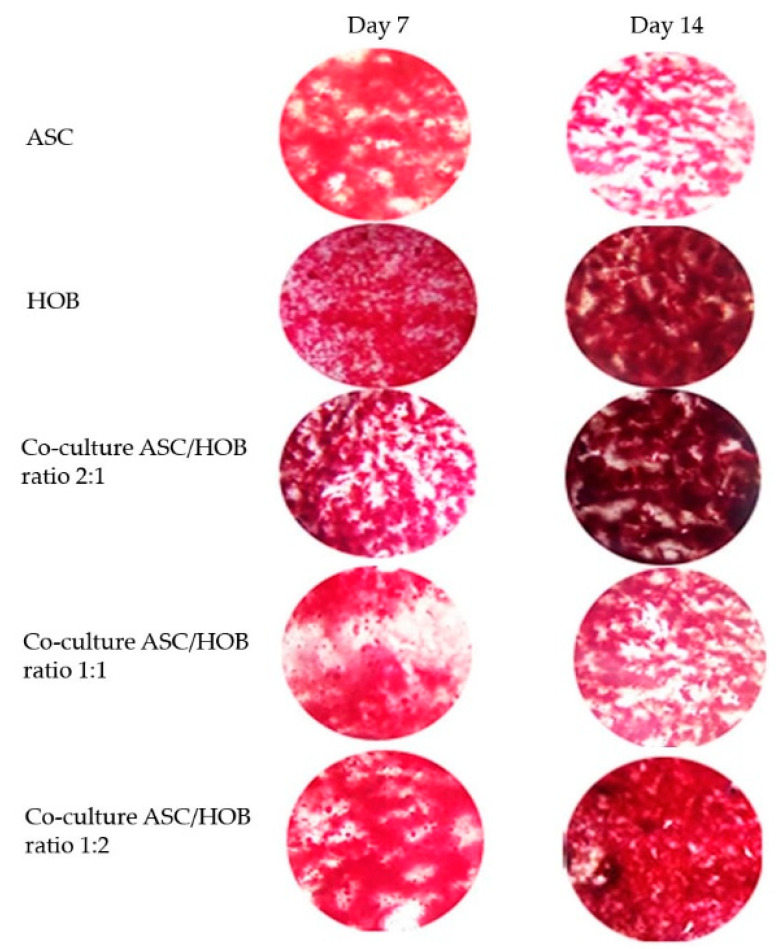
Representative images of cells from all groups seeded in PCL-HA stained with Alizarin Red S at days 7 and 14. Significant difference can be observed at day 14 for most groups, particularly for ASC/HOB co-cultured at 2:1.

**Figure 6 polymers-13-00597-f006:**
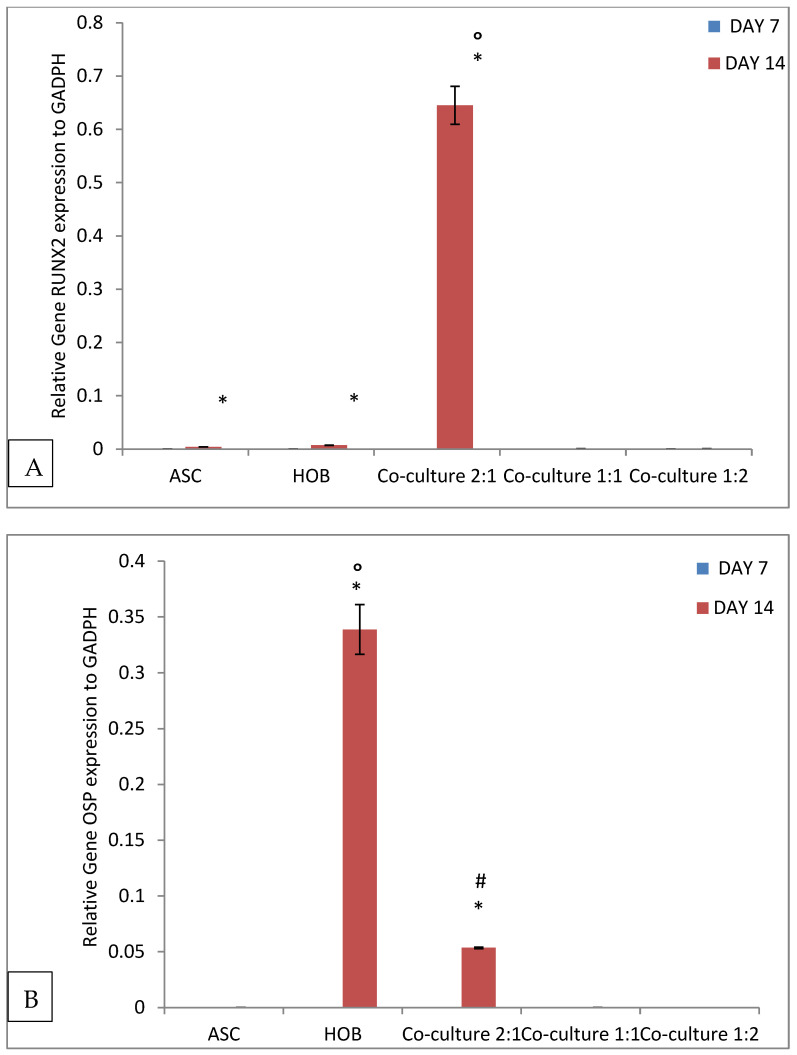
Osteogenic gene expression levels of cells seeded in PCL-HA at days 7 and 14: (**A**) Runx; (**B**) OSP; (**C**) ALP; (**D**) BSP and (**E**) OCN, with day 14 showed significant changes in the expression level, particularly for co-cultured ASC/HOB at 2:1 ratio. * Indicates significant difference of *p* < 0.05, relative to day 7, O indicates significant difference of *p* < 0.05 relative to all groups, # indicates significant difference relative to other co-culture groups, ASC/HOB at 1:1 and 1:2.

**Table 1 polymers-13-00597-t001:** Description of osteogenic associated gene primers used in RT-PCR.

Gene	Accession No.	Primer (5′ → 3′) Sense and Antisense	PCR Product Size (bp)
Glyceraldehyde-3-phosphate dehydrogenase (GADPH)	NM_002046	R 5′-TCC CTG AGC TGA ACG GGA AG-3′	217
		F 5′-GGA GGA GTG GGT GTC GCT GT-3′	
Osteopontin (OSP)	NM_001040060	R 5′-ATCCATGTGGTCATGGCTTT-3′	219
		F 5′-CACCTGTGCCATACCAGTTAAAC-3′	
Osteocalcin (OCN)	NM_199173	R 5′-CTGAAAGCCGATGTGGTCAG-3′	191
		F 5′-GTGCAGAGTCCAGCAAAGGT-3′	
Bone-sialoprotein (BSP)	NM_004967	R 5′-CTCGGTAATTGTCCCCACGA-3′	208
		F 5′-GGGCACCTCGAAGACAACAA-3′	
Runt-2 (RUNX)	NM_004348	R 5′-CACTCTGGCTTTGGGAAGAG-3′	182
		F 5′-GCAGTTCCCAAGCATTTCATC-3′	
alkaline phosphatase (ALP)	NM_000478	R 5′-AGGGGAACTTGTCCATCTCC-3′	200
		F 5′-GTACTGGCGAGACCAAGCGCA-3′	

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
