# Peer review of "Polycaprolactone-Based Scaffolds Facilitates Osteogenic Differentiation of Human Adipose-Derived Stem Cells in a Co-Culture System"

_polymers, 2021, doi:10.3390/polym13040597_

Round 1

Reviewer 1 Report

In this manuscript, authors have investigated an osteogenic differentiation of human adipose stem cells co-cultured with human osteoblasts on polycaprolactone/hydroxyapatite-based scaffold. Although there is no much novelty in terms of composite biomaterial, but co-culturing of cells on this biomaterial is interesting. This material is characterized and well presented. However, there are two major concerns for improving this manuscript.

  1. Alldiagram figures are not plotted well and not in good resolution. Figures should be provided by good professional software for better presentation of the results.
  2. Qualitativeanalysis (e.g. live/dead assay or other staining images) should be incorporated in this study.  

Author Response

Thank you for the valuable comments. Our responses are in the attachment.

Reviewer 2 Report

The current manuscript is answering a question regarding an emerging issue in the realm of bone tissue engineering. The authors could successfully hypothesize the design of a sophisticated scaffold based on HA and PCL. The results are promising when it comes to osteogenic differentiation of adipose-derived stem cells. However, there are some changes that the authors are strongly encouraged to implement in the manuscript:

1- The quality of the images is very low and it is not well designed. Figure 5 should be corrected. Figure 5 is very disorganized. These figures should be submitted to the journal a raw format.

2- This article is talking about osteogenic differentiation of stem cells. However the authors fail to explain the effect of HA such differentiation. These two articles is suggested to be cited in the introduction where the authors mention “Scaffold materials such as hydroxyapatite (HA) have been shown to aid stem cells in osteoblast differentiation”:

1- Barradas AM, Monticone V, Hulsman M, Danoux C, Fernandes H, Tahmasebi Birgani Z, Barrère-de Groot F, Yuan H, Reinders M, Habibovic P, van Blitterswijk C, de Boer J. Molecular mechanisms of biomaterial-driven osteogenic differentiation in human mesenchymal stromal cells. Integr Biol (Camb). 2013 Jul 24;5(7):920-31. doi: 10.1039/c3ib40027a. Epub 2013 Jun 11. PMID: 23752904.

2- Hesaraki S, Nazarian H, Pourbaghi-Masouleh M, Borhan S. Comparative study of mesenchymal stem cells osteogenic differentiation on low-temperature biomineralized nanocrystalline carbonated hydroxyapatite and sintered hydroxyapatite. J Biomed Mater Res B Appl Biomater. 2014 Jan;102(1):108-18. doi: 10.1002/jbm.b.32987. Epub 2013 Jul 13. PMID: 23853054.

The authors should explain why the calcium release into the microenvironment can benefit the osteogenic differentiation.

3- The caption for Figure 1 should emphasize about the technique used for imaging (SEM).

The authors are encouraged to resubmit the revised manuscript as soon as possible.

Regards

Author Response

Thank you for your valuable comments, we have response to you comment as in the attachment.

Round 2

Reviewer 1 Report

In my opinion, this manuscript can now be accepted for publication.

Reviewer 2 Report

To the authors,

Thank you for revising the manuscript and implementing the comments in the manuscript. I believe that the quality of this manuscript is good and could interesting for the readers.

Regards